# Is Biportal Endoscopic Spine Surgery More Advantageous Than Uniportal for the Treatment of Lumbar Degenerative Disease? A Meta-Analysis

**DOI:** 10.3390/medicina58111523

**Published:** 2022-10-26

**Authors:** Wen-Bin Xu, Vit Kotheeranurak, Huang-Lin Zhang, Zhang-Xin Chen, Hua-Jian Wu, Chien-Min Chen, Guang-Xun Lin, Gang Rui

**Affiliations:** 1Department of Orthopedics, The First Affiliated Hospital of Xiamen University, School of Medicine, Xiamen University, Xiamen 361000, China; 2Department of Orthopedics, Faculty of Medicine, Chulalongkorn University, and King Chulalongkorn Memorial Hospital, Bangkok 10330, Thailand; 3Center of Excellence in Biomechanics and Innovative Spine Surgery, Chulalongkorn University, Bangkok 10330, Thailand; 4The Third Clinical Medical College, Fujian Medical University, Fuzhou 350122, China; 5Department of Orthopedics, Affiliated Dongnan Hospital of Xiamen University, Xiamen University, Zhangzhou 363000, China; 6Division of Neurosurgery, Department of Surgery, Changhua Christian Hospital, Changhua 500209, Taiwan; 7Department of Leisure Industry Management, National Chin-Yi University of Technology, Taichung 41170, Taiwan; 8School of Medicine, Kaohsiung Medical University, Kaohsiung 80708, Taiwan

**Keywords:** lumbar degenerative disease, uniportal endoscopic, biportal endoscopic, lumbar decompression laminectomy, meta-analysis

## Abstract

*Background and Objectives*: To estimate the clinical outcomes of uniportal and biportal full-endoscopic spine surgery for the treatment of lumbar degenerative disease (LDD), and to provide the latest evidence for clinical selection. *Materials and Methods*: Relevant literatures published in PubMed, Web of Science, Embase, CNKI, and WanFang Database before 21 November 2021 were searched systematically. Two researchers independently screened the studies, extracted data, and evaluated the risk of bias of the included studies. The systematic review and meta-analysis were performed using the Review Manager software (version 5.4; The Cochrane Collaboration). *Results*: A total of seven studies were included in this meta-analysis, including 198 patients in a uniportal endoscopy group and 185 patients in a biportal endoscopy group. The results of this meta-analysis demonstrated that the biportal endoscopy group experienced less intraoperative estimated blood loss (WMD = −2.54, 95%CI [−4.48, −0.60], *p* = 0.01), while the uniportal endoscopy group displayed significantly better recovery results in Visual Analog Scale (VAS) assessments of the back within 3 days of surgery (WMD = 0.69, 95%CI [0.02, 1.37], *p* = 0.04). However, no significant differences in operation time, length of hospital stay, complication rates, Oswestry Disability Index (ODI) (within 3 months), ODI (last follow-up), VAS for back (within 3 months), VAS for back (last follow-up), and VAS for leg (within 3 days, within 3 months, last follow-up) were identified between the two groups. *Conclusions*: According to our meta-analysis, patients who underwent the uniportal endoscopic procedure had more significant early postoperative back pain relief than those who underwent the biportal endoscopic procedure. Nevertheless, both surgical techniques are safe and effective.

## 1. Introduction

Lumbar degenerative disease (LDD) is a common disease caused by natural aging of the lumbar spine with clinical symptoms such as lumbar disc herniation (LDH), lumbar spinal stenosis (LSS), and lumbar spondylolisthesis. With the progression of global aging, the incidence of LDD is gradually increasing [1]. The vast majority of patients with LDD suffer from low back pain, conduction pain of the lower limbs, muscle weakness, and other symptoms. Surgical intervention is required when the symptoms are not relieved or are worsened by long-term conservative treatments [2].

Following decades of progress, full-endoscopic spine surgery is gaining popularity among patients and has emerged as a hotspot and new direction for spine surgery. This technique has several advantages, such as fewer complications, faster postoperative recovery, and less damage to bony structures and soft tissues [3]. Furthermore, the uniportal endoscopy system has several disadvantages such as its lack of flexibility and low efficiency of tissue processing [4]. Recently, the unilateral biportal endoscopic (UBE) spine surgery technique has received considerable attention due to its relatively gentle learning curve, wide endoscopic view, and greater flexibility in instrumentation, especially in the decompression of the spinal canal, which is more efficient [5,6,7,8]. This surgical method establishes two channels: the visualization channel for endoscope placement and the working channel for instrument operation [9]. This approach combines the merits of traditional uniportal endoscopic spinal surgery and open surgery [10].

In recent years, several studies have compared uniportal and biportal endoscopic spine surgery procedures for the treatment of LDD [11,12,13,14,15,16,17], but no meta-analysis directly comparing the two techniques has been published to date. Here, we performed a meta-analysis to examine the most recent evidence and compare these two approaches.

## 2. Materials and Methods

### 2.1. Surgical Technique of Biportal Endoscopy

The biportal endoscopy procedure is usually performed under general or epidural anesthesia with the patient in the prone position. After induction of anesthesia, the level of the target lesion is confirmed under C-arm fluoroscopy.

The biportal endoscopy requires the creation of two channels: an endoscopic channel (visual portal) and an instrumental channel (working portal). In general, the working portal is created on the disc level, followed by the endoscope portal, which is created 2.0–3.0 cm apart from the working portal. The surgical incision can be altered depending on the type of disc herniation, the severity of stenosis, or the presence of obesity. As a result, the working portal is identified first, and is then followed by the endoscope portal. To avoid interruption, the two portals should be at least 2.0–3.0 cm apart. Transverse or longitudinal incisions are feasible. The paravertebral muscles are dilated layer by layer, using a dilator to gently push the soft tissue apart and create a workspace. With the inflow of saline, the space is formed and prepared for use. After confirming the inferior margin of the superior lamina, a laminectomy is performed with a drill starting from the inferior margin of the superior lamina and continuing until the superior margin of the ligamentum flavum (LF) is exposed. Drilling the central portion first rather than the lateral portion may help to minimize excessive facet joint resection at first. The laminectomy is continued downward to expose the origin of the LF and the underlying epidural fat. The LF is removed, and the lateral margins of the dural sac and nerve roots are identified. After ensuring that the nerve roots are protected with retractors, the discectomy is performed with forceps.

In patients with spinal stenosis, the lower surface of the contralateral lamina is abraded until the lateral recess is approached, and bilateral decompression is performed in a unilateral manner. In this case, the LF must be used as a neuroprotective tissue, and it is recommended that the flavectomy be performed after bony decompression. In addition, in cases of severe stenosis, the spinous process and facet joint are usually hypertrophied and deformed. In such an instance, a contralateral laminotomy should be performed from the base of the spinous process to make it easier to reach the contralateral lateral recess. After adequate decompression, the skin incisions are closed and a drainage tube is placed as appropriate.

### 2.2. Literature Search

The PubMed (MEDLINE), Web of Science, Embase, CNKI, and WanFang databases were searched, according to the Preferred Reporting Items for Systematic Reviews and Meta-Analyses (PRISMA) guidelines and Cochrane Collaboration recommendations, to find all related articles published before 21 November 2021 [18,19,20]. The search terms were as follows: “biportal endoscopy*”, “biportal endoscopic*”, “two portal endoscopy*”, “two portal endoscopic*”, “irrigation endoscopy*”, “irrigation endoscopic*”, “uniportal endoscopy*”, “uniportal endoscopic*”, “percutaneous endoscopy*”, “percutaneous endoscopic*”, “percutaneous transforaminal endoscopy*”, “percutaneous transforaminal endoscopic*”, “percutaneous interlaminar endoscopy*”, “percutaneous interlaminar endoscopic*”, “Yeung endoscopic spine system,” “YESS,” “transforaminal endoscopic spine system,” “TESSYS,” “full endoscopy*”, and “full endoscopic*”. Articles written in English and Chinese were included in the study.

### 2.3. Inclusion Criteria

(1)Research type: randomized controlled trials (RCTs) and retrospective/prospective cohort or case–control studies(2)Patients with lumbar degenerative disease (LDH/LSS) with indications for surgery(3)Comparison of clinical results between uniportal endoscopic and biportal endoscopic decompression in treating lumbar degenerative disease (LDH/LSS)(4)Articles with at least one of the following results: primary results including pain intensity (Visual Analog Scale [VAS] scores) and disability (Oswestry Disability Index [ODI] scores), or secondary results including operation time, intraoperative estimated blood loss, length of hospital stay, and complications.

### 2.4. Exclusion Criteria

(1)Non-English and non-Chinese studies(2)Studies that have not been peer-reviewed or those without relevant data, such as case series, conference reports, letters, and reviews(3)Repeatedly published data(4)Patients with serious cardiovascular and cerebrovascular diseases, mental diseases, malignant tumors, etc.

### 2.5. Study Selection

Two researchers (W.-B.X. and G.-X.L.) independently screened all relevant articles in accordance with the inclusion and exclusion criteria. In the process of study selection, disagreements between the two researchers were resolved by discussion or with the help of a third-party researcher (R.G.).

### 2.6. Risk of Bias within Included Articles

Randomized controlled trials were analyzed for risk of bias using the criteria recommended by the Collaboration’s tool [21]. The risk of bias of the cohort or case–control studies was assessed using the Newcastle–Ottawa scale (NOS) [22].

### 2.7. Data Extraction

Two researchers (W.-B.X. and G.-X.L.) independently extracted the data according to a standardized form established separately for this meta-analysis. The extracted data included study time, country, study design, interventions, number of patients, age, sex, mean follow-up time, and outcomes (number of complications, pain intensity, disability, operation time, intraoperative estimated blood loss, and length of hospital stay).

### 2.8. Data Analysis

Meta-analysis was performed using Review Manager software (version 5.4; The Cochrane Collaboration). Odds ratios (ORs) were estimated for dichotomous variables, and weighted mean differences (WMD) were utilized for continuous variables with 95% confidence intervals (CIs) for each outcome. Statistical significance was set at *p* < 0.05. The chi-squared (*I*^2^) statistic [23] was used to measure heterogeneity among the included trials. An *I*^2^ value of >50% or *p* ≤ 0.10 indicated substantial heterogeneity, and a random-effects model was used to compare results with heterogeneity; otherwise, a fixed-effects model was used. Subgroup analysis was used to investigate the potential source of heterogeneity in the studies, and sensitivity analysis was performed by observing the change in pooled effect size after removing included studies one by one. Publication bias of the included studies was evaluated using funnel plots.

## 3. Results

### 3.1. Description of Articles

The screening process and outcomes are shown in the PRISMA flow chart (Figure 1). Based on our initial literature search, 726 papers were included. After abstract review, duplicates and papers that did not meet the inclusion criteria were excluded, and eight relevant articles were obtained. The selected relevant studies were read in full text, and seven articles were eventually included in this meta-analysis, including three English studies [15,16,17] and four Chinese studies [11,12,13,14].

A total of seven studies that enrolled 383 patients (uniportal endoscopy group: 198 cases; biportal endoscopy group: 185 cases) met the inclusion criteria. Of the seven articles, one study [13] was a randomized controlled trial, one study [15] was a prospective study, and five studies [11,12,14,16,17] were retrospective studies. In terms of the different disease types of patients, three articles [11,12,16] involved patients with LSS, and four articles [13,14,15,17] involved patients with LDH. All included studies were published in complete manuscript form. Table 1 shows the concrete baseline information for the seven articles.

### 3.2. Quality Evaluation

The quality of the randomized controlled study was described in accordance with the Cochrane Collaboration’s tool (Figure 2). The quality of non-randomized articles was assessed using the NOS (Table 2). Six articles were non-randomized studies with a NOS score ≥ 6 points (low risk of bias) [11,12,14,15,16,17]. In summary, the quality of the included studies was moderate to high.

### 3.3. Meta-Analysis Results

#### 3.3.1. VAS for Back (within 3 Days)

The level of pain intensity within the follow-up period was available from seven articles [11,12,13,14,15,16,17]; however, the VAS scores of the back and leg were merged in three articles [11,12,13]; therefore, we excluded these studies to avoid causing significant heterogeneity.

Four studies [14,15,16,17] reported postoperative VAS scores for back pain (within 3 days) in the uniportal and biportal endoscopy groups. A total of 197 patients (uniportal endoscopy group: 105 cases; biportal endoscopy group: 92 cases) were included. The analysis demonstrated great significance in heterogeneity (*I*^2^ = 92%, *p* < 0.1); hence, a random-effects model was applied. Meta-analysis showed that the uniportal endoscopy group studies reported lower VAS scores for back pain (within 3 days) than the biportal endoscopy group, with statistically significant differences (WMD = 0.69, 95%CI [0.02, 1.37], *p* = 0.04) (Figure 3A). Based on this analysis, four studies were divided into the LSS and LDH subgroups according to the different disease types of patients.

Subgroup analysis indicated that: (1) heterogeneity between the LSS subgroup and the LDH subgroup was extremely high (*I*^2^ = 79%, *p* = 0.03), which indicates that the different disease types of patients would greatly affect the results of this meta-analysis; (2) the LDH subgroup involving three studies [14,15,17] also suggested that the uniportal endoscopy group had lower VAS scores for back pain (within 3 days) than the biportal endoscopy group, with statistically significant differences [WMD = 0.91, 95%CI (0.22, 1.61), *p* = 0.01] (Figure 3A); and (3) the LSS subgroup involving one study [16] suggested that there were no differences between the groups (WMD = 0.04, 95%CI [−0.33, 0.41], *p* = 0.83) (Figure 3A).

#### 3.3.2. VAS for Back (within 3 Months)

Two studies [15,17] reported postoperative VAS scores for back pain (within 3 months) in the uniportal and biportal endoscopy groups. A total of 100 patients (uniportal endoscopy group: 60 cases; biportal endoscopy group, 40 cases) were included. The analysis demonstrated no significant heterogeneity between these articles (*I*^2^ = 0%, *p* = 0.77); hence, a fixed-effects model was applied. Meta-analysis showed that the VAS scores for back pain (within 3 days) were similar for both groups (WMD = 0.07, 95%CI [−0.16, 0.29], *p* = 0.56) (Figure 3B).

#### 3.3.3. VAS for Back (Last Follow-Up)

Four studies [14,15,16,17] reported postoperative VAS scores for back pain (last follow-up) in the uniportal and biportal endoscopy groups. A total of 197 patients (uniportal endoscopy group: 105 cases; biportal endoscopy group: 92 cases) were included. The analysis demonstrated great significance in heterogeneity (*I*^2^ = 94%, *p* < 0.1); hence, a random-effects model was applied. Meta-analysis showed that VAS scores for back pain (last follow-up) were similar between the two groups (WMD = 0.46, 95%CI [−0.16, 1.08], *p* = 0.14) (Figure 3C). Based on this analysis, four studies were divided into the sample size ≥ 20 subgroups and the sample size < 20 subgroups, depending on the sample size of the patients.

Subgroup analysis indicated that: (1) heterogeneity between the sample size ≥ 20 subgroup and the sample size < 20 subgroup was extremely high (*I*^2^ = 98.1%, *p* < 0.1), which indicates that the different patient sample sizes would greatly affect the results of this meta-analysis; (2) sample size ≥ 20 subgroup involving three studies [15,16,17] suggested that back pain relief at last follow-up was similar in both groups (WMD = 0.10, 95%CI [−0.05, 0.24], *p* = 0.21) (Figure 3D); and (3) sample size < 20 subgroup involving one study [14] suggested that there were significant differences (WMD = 1.60, 95%CI [1.22, 1.98], *p* < 0.05) (Figure 3D).

#### 3.3.4. VAS for Leg

Four articles [14,15,16,17] reported postoperative VAS scores for leg pain (within 3 days) in the uniportal and biportal endoscopy groups. A total of 197 patients (uniportal endoscopy group: 105 cases; biportal endoscopy group: 92 cases) were included. The analysis demonstrated no significant heterogeneity between these articles (*I*^2^ = 25%, *p* = 0.26); hence, a fixed-effects model was applied. Meta-analysis showed that VAS scores for leg pain (within 3 days) were similar in both groups (WMD = 0.04, 95%CI [−0.25, 0.33], *p* = 0.78) (Figure 4A). 

Two studies [15,17] reported postoperative VAS scores for leg pain (within 3 months) in the uniportal and biportal endoscopy groups. A total of 100 patients (uniportal endoscopy group: 60 cases; biportal endoscopy group: 40 cases) were included. The analysis demonstrated no significant heterogeneity between these articles (*I*^2^ = 0%, *p* = 0.63); hence, a fixed-effects model was applied. Meta-analysis showed that VAS scores for leg pain (within 3 months) were similar in both groups (WMD = 0.23, 95%CI [−0.03, 0.50], *p* = 0.09) (Figure 4B).

Four articles [14,15,16,17] reported postoperative VAS scores for leg pain (last follow-up) in the uniportal and biportal endoscopy groups. A total of 197 patients (uniportal endoscopy group: 105 cases; biportal endoscopy group: 92 cases) were included. The analysis demonstrated no significant heterogeneity between these articles (*I*^2^ = 0%, *p* = 0.91); hence, a fixed-effects model was applied. Meta-analysis showed that VAS scores for leg pain (last follow-up) were similar in both groups (WMD = 0.13, 95%CI [0.00, 0.27], *p* = 0.05) (Figure 4C).

#### 3.3.5. ODI

Six articles [11,12,13,14,15,17] reported postoperative ODI scores (within 3 months) in the uniportal and biportal endoscopy groups. A total of 319 patients (uniportal endoscopy group: 171 cases; biportal endoscopy group: 148 cases) were included. The analysis demonstrated that there was mild heterogeneity between these articles (*I*^2^ = 64%, *p* = 0.02); hence, a random-effects model was applied. Meta-analysis showed that ODI scores (within 3 months) were similar between the two groups (WMD = −0.33, 95%CI [−1.85, 1.18], *p* = 0.67) (Figure 5A).

Seven studies [11,12,13,14,15,16,17] reported postoperative ODI scores (last follow-up) of the uniportal and biportal endoscopy groups. A total of 383 patients (uniportal endoscopy group: 198 cases; biportal endoscopy group: 185 cases) were included. The analysis demonstrated there was mild heterogeneity between these articles (*I*^2^ = 52%, *p* = 0.05); hence, a random-effects model was applied. Meta-analysis showed that ODI scores (last follow-up) were similar in both groups (WMD = −0.53, 95%CI [−1.36, 0.29], *p* = 0.21) (Figure 5B).

#### 3.3.6. Operation Time

Seven articles [11,12,13,14,15,16,17] reported the operation time of the uniportal and biportal endoscopy groups. A total of 383 patients (uniportal endoscopy group: 198 cases; biportal endoscopy group: 185 cases) were included. The analysis demonstrated great significance in heterogeneity (*I*^2^ = 99%, *p* < 0.1), and sensitivity analysis was performed to explore the source of heterogeneity. First, the pooled papers were excluded one by one, and the remaining studies were pooled again, showing that the heterogeneity of each group remained high. Second, we found that heterogeneity remained high after changing the effect model. Therefore, sensitivity analysis suggested that the results were reliable, and the source of heterogeneity might be related to the techniques of different operators. Meta-analysis showed that the mean operation time was similar between the two groups (WMD = 11.75, 95%CI [−4.35, 27.84], *p* = 0.15) (Figure 6A).

#### 3.3.7. Length of Hospital Stay

Five articles [11,12,14,15,17] reported the postoperative length of hospital stay in the uniportal and biportal endoscopy groups. A total of 281 patients (uniportal endoscopy group: 153 cases; biportal endoscopy group: 128 cases) were included. The analysis demonstrated great significance in heterogeneity (*I*^2^ = 98%, *p* < 0.1), and sensitivity analysis was used to determine the origin of heterogeneity. We found that Hao 2021 [17] had the highest heterogeneity. After removing this article, the analysis indicated no significant heterogeneity among the remaining four articles (*I*^2^ = 44%, *p* = 0.15); thus, a fixed-effects model was used. Meta-analysis showed that the length of hospital stay was similar between the two groups (WMD = −0.20, 95%CI [−0.56, 0.16], *p* = 0.28) (Figure 6B).

#### 3.3.8. Complications

Seven articles [11,12,13,14,15,16,17] reported complication rates in the uniportal and biportal endoscopy groups. A total of 383 patients (uniportal endoscopy group, 198 patients; biportal endoscopy group, 185 cases) were included. The analysis demonstrated no significant heterogeneity between these articles (*I*^2^ = 0%, *p* = 0.80); hence, a fixed-effects model was applied. Meta-analysis showed that complication rates were similar in both groups (OR = 0.86, 95%CI [0.36, 2.02], *p* = 0.72) (Figure 6C). Details of the complications are shown in Table 3.

#### 3.3.9. Intraoperative Estimated Blood Loss

Three articles [11,12,17] reported intraoperative estimated blood loss in the uniportal and biportal endoscopy groups. A total of 188 patients (uniportal endoscopy group: 95 cases; biportal endoscopy group, 93 cases) were included. The analysis demonstrated great significance in heterogeneity (*I*^2^ = 98%, *p* < 0.1), and sensitivity analysis was used to determine the origin of heterogeneity. We found that Hao 2021 [17] had the highest heterogeneity. After removing this article, the analysis indicated no significant heterogeneity among the remaining two articles (*I*^2^ = 0%, *p* = 0.65); thus, a fixed-effects model was used. Meta-analysis showed less intraoperative estimated blood loss was found in the biportal endoscopy group (WMD = −2.54, 95%CI [−4.48, −0.60], *p* = 0.01) (Figure 6D).

### 3.4. Sensitivity Analysis

A sensitivity analysis was required to examine the stability of the results. The analysis revealed that operation time, intraoperative estimated blood loss, length of hospital stay, ODI (within 3 months), ODI (last follow-up), VAS score for back (within 3 days), and VAS score for back (last follow-up) showed significant heterogeneity.

For operation time, the included studies were excluded one by one, and the remaining articles were pooled. Sensitivity analysis demonstrated that the heterogeneity remained high, indicating that the results were relatively stable, and the heterogeneity may be related to the surgeons’ surgical skill level. 

For intraoperative estimated blood loss, sensitivity analysis revealed that the heterogeneity of Hao 2021 [17] was the highest. After removing this article, heterogeneity decreased from *I*^2^ = 98% to *I*^2^ = 0%, indicating that the heterogeneity mainly came from Hao 2021 [17]. A forest plot without Hao’s article is shown in Figure 6D. 

Regarding the length of hospital stay, sensitivity analysis revealed that the heterogeneity of Hao 2021 [17] was the highest. After removing this article, heterogeneity decreased from *I*^2^ = 93% to *I*^2^ = 46%, indicating that the heterogeneity mainly came from Hao 2021 [17]. A forest plot without Hao’s article is shown in Figure 6B. 

For ODI, there was mild heterogeneity (within 3 months: *I*^2^ = 64%; last follow-up: *I*^2^ = 52%). After excluding the studies one by one, we found that the meta-analysis results did not change, indicating that the results were relatively stable. Therefore, a random-effects model was used.

For back pain VAS (within 3 days; final follow-up), subgroup analysis was performed to find heterogeneity.

### 3.5. Publication Bias

Funnel plots (length of hospital stay; VAS for back [within 3 days]) were analyzed, and the results showed that the funnel plots were symmetrical (Figure 7).

## 4. Discussion

The results of this study reveal that both uniportal endoscopy and biportal endoscopy are safe and effective procedures for treating LDD. Biportal endoscopy is associated with less intraoperative estimated blood loss, while uniportal endoscopy is associated with early back pain relief. No significant differences in operation time, length of hospital stay, complication rates, ODI (within 3 months), ODI (last follow-up), VAS for back (within 3 months), VAS for back (last follow-up), and VAS for leg (within 3 days, within 3 months, last follow-up) were identified between the two groups. Notably, the analysis of the severity of LDD was not performed in this study, so it is unclear whether the severity of the disease may influence the outcomes of treatment.

The primary purpose of treatment for LDD is to alleviate clinical symptoms and improve patient prognosis. Uniportal endoscopy is characterized by fewer complications, faster postoperative recovery, and reduced damage [3]. Biportal endoscopy is an emerging method that combines the advantages of microscopy technology and uniportal endoscopy technology. These include independent visualization and manipulation channels, a flexible instrument-manipulation space, a clear and wide surgical field of vision, and the use of conventional open-spinal surgery instruments. However, prior to our study, there were no meta-analyses that provided evidence for comparing the two techniques to determine which is more clinically effective in the treatment of LDD. Therefore, we performed this meta-analysis to comprehensively evaluate the correlation between the two techniques and the prognosis of patients with LDD. The results will assist and guide clinical decision-making.

### 4.1. Perioperative Data

The operation time and length of hospital stay are similar between uniportal and biportal endoscopy. However, less bleeding occurs following biportal endoscopy compared to uniportal endoscopy. A possible explanation is that biportal endoscopy has sufficient space, as well as a good surgical field of view, to allow for accurate hemostasis and to avoid failure in finding the bleeding point due to the limited operating space and restricted field of view. In addition, the surgical approach is similar to traditional open surgery, allowing for better identification of anatomical structures and thus minimizing unnecessary injuries [24,25,26,27].

### 4.2. VAS Score and ODI

Regarding postoperative evaluation indices, the VAS scores for back pain within 3 days after surgery in the uniportal endoscopy group were better than those in the biportal endoscopy group.

In the percutaneous transforaminal endoscopic discectomy (PTED) technique, most of the posterior structures such as the LF are preserved [28,29], rather than being removed as they are with the biportal endoscopy technique [30]. Some articles reported that the remaining LF can prevent scar formation and improve clinical efficacy [31,32]. Back pain may also be associated with muscle and soft tissue injuries caused by preparation of the workspace. Compared to the percutaneous interlaminar endoscopic discectomy (PIED) technique, the biportal endoscopy technique requires a larger space for manipulation, resulting in damage to the surrounding muscles and soft tissues. All of the above suggest that uniportal endoscopy is more effective than biportal endoscopy for early back pain relief. Subgroup analysis showed that in terms of early postoperative back pain relief (within 3 days), uniportal endoscopy had an advantage over biportal endoscopy in the treatment of patients with LDH, while no such difference was found in the LSS subgroup. We believe that the primary reasons for this difference are as follows. First, for patients with LDH, the biportal endoscopy technique includes two portals, one of which is the working portal where surgical instruments are inserted and removed repeatedly. Therefore, any injury generated by the biportal endoscopy technique may be more serious than uniportal endoscopy, resulting in better early postoperative back pain recovery after uniportal endoscopy than biportal endoscopy. Second, for patients with LSS, due to the flexibility of the operation and the routine use of open surgery instruments, the biportal endoscopy technique has an advantage in laminectomy, especially in bilateral decompression, in terms of operation time. However, a senior endoscopic surgeon can also perform laminectomy quickly and effectively using the uniportal endoscopic technique. This may explain why no statistically significant difference in operation time was found in our analysis.

In our meta-analysis, no noteworthy differences were detected among groups with regard to VAS for back (within 3 months), VAS for back (last follow-up), and VAS for leg (within 3 days, within 3 months, last follow-up). However, the sample size < 20 subgroup had better back pain relief in the postoperative period (last follow-up) in the uniportal endoscopy group, which is likely due to bias caused by the inclusion of only one study and the study having a small sample size. Therefore, readers should be cautious when interpreting this result.

ODI is considered as one of the main outcome measures that is broadly applied in the assessment of patients with spinal diseases. ODI is prominently associated with pain scores [33]. This meta-analysis revealed that whether uniportal endoscopy or biportal endoscopy was used to treat LDD, no significant differences were detected at ODI within 3 months and ODI at last follow-up, indicating that the efficacy of the two surgical methods was similar.

### 4.3. Complication

Complications are one of the key issues that surgeons and patients must consider. Our data demonstrated that eleven (5.95%) of the 185 patients experienced complications associated with biportal endoscopy, and twelve (6.06%) of the 198 patients developed complications associated with uniportal endoscopy. Although there was no statistical difference between the two groups, the uniportal endoscopy group had a slightly higher incidence of complications than the biportal endoscopy group. One of the main reasons for these findings may be that biportal endoscopy expands the surgical field of view and reduces the difficulty of surgery, thus reducing the occurrence of potential surgical complications. The common complications of the biportal endoscopy technique were dural tears (n = 4), nerve root injury (n = 3), cerebrospinal fluid leakage (n = 2), infection (n = 2), postoperative hematoma (n = 1), and transient paresthesia (n = 1). The vast majority of these dural tears and nerve root injuries were attributed to intraoperative use of the drill and Kerrison Rongeur. In addition, the common complications of uniportal endoscopy were dural tears (n = 5), infection (n = 4), nerve root injury (n = 2), transient paresthesia (n = 2), transient weakness (n = 1), and postoperative hematoma (n = 1). It is known that the PTED technique involves entrance from the intervertebral foramen and that the surgeon does not directly pull the nerve root and the dural sac, which is instrumental in avoiding dural tears and nerve root injury [34,35]. However, when central LDH is treated with the PIED technique, it is usually to access the nerve root by increasing the inclination angle of the working cannula or rotating the working cannula, which increases the risk of dural tears and ‘overpulling’ of the nerve root [34]. Therefore, the PIED technique is more likely to lead to complications as a result of dural tears and nerve root injury than the PTED technique. In addition, postoperative hematomas have been reported in both types of surgical procedures. The majority of these postoperative hematomas are asymptomatic and recover gradually with conservative treatment.

### 4.4. Limitations and Strength

In this study, a few limitations need to be mentioned: (1) the inclusion of one RCT, one prospective study, and five retrospective studies somewhat lower the level of evidence-based medicine in this meta-analysis; (2) the number of included studies is limited, and higher quality, larger sample size clinical trials are required to support the findings of the meta-analysis; (3) some of the included studies are only descriptive and cannot provide accurate clinical data; and (4) the technical level of different surgeons may have influenced the results; (5) the age of patients in included studies is relatively young, and clinical studies involving more elderly patients need to be included to validate the results of this meta-analysis. Despite these limitations, we consider this to be the first meta-analysis on this subject that contains whole clinical trials. Through an overall literature search, independent data collection, and rigorous inclusion and exclusion criteria, potential bias was minimized. 

## 5. Conclusions

In summary, our findings show that despite differences in technique, both uniportal endoscopy and biportal endoscopy are safe and effective in treating LDD. Furthermore, patients who underwent the uniportal endoscopic procedure had more significant early postoperative back pain relief than those who underwent the biportal endoscopic procedure. Restricted by the quantity of included papers, these findings need to be further verified by more high-quality prospective RCTs.

## Figures and Tables

**Figure 1 medicina-58-01523-f001:**
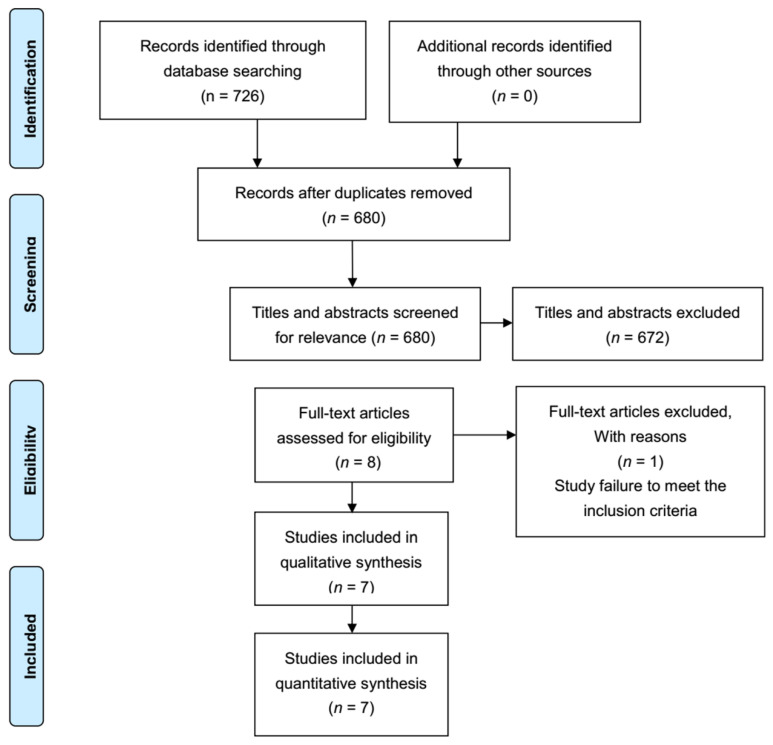
PRISMA flow chart of literature search.

**Figure 2 medicina-58-01523-f002:**
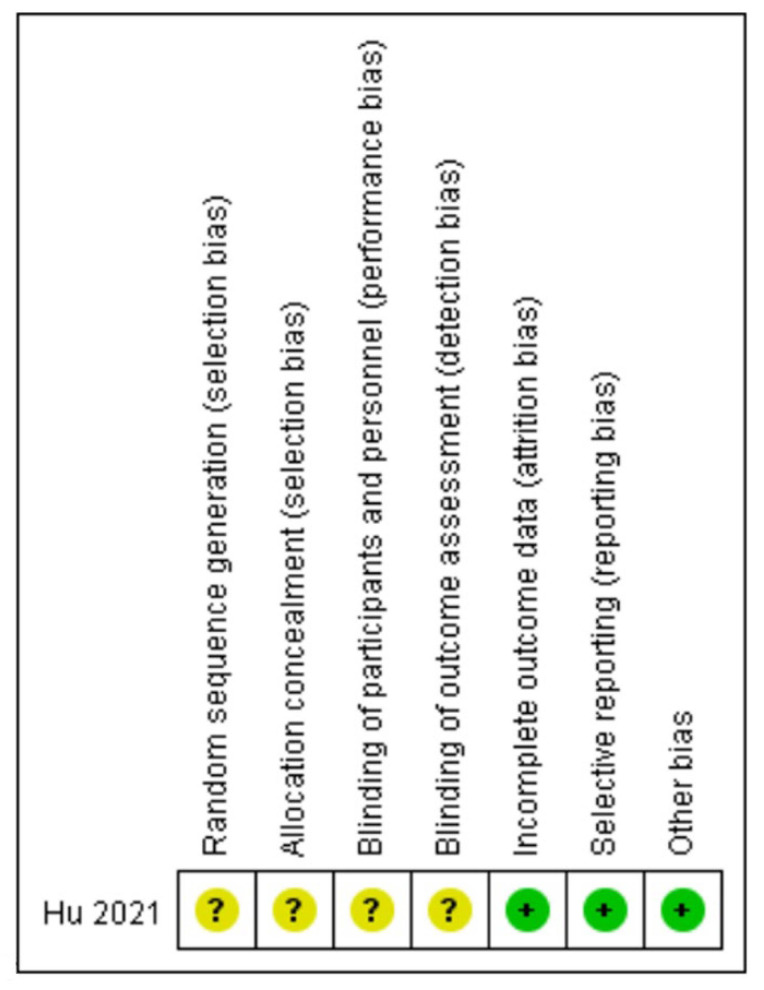
Risk of bias summary: review authors’ judgement regarding each risk-of-bias item for RCT. Green round represents low risk and yellow round represents not mentioned in the article Hu et al., 2021 [13].

**Figure 3 medicina-58-01523-f003:**
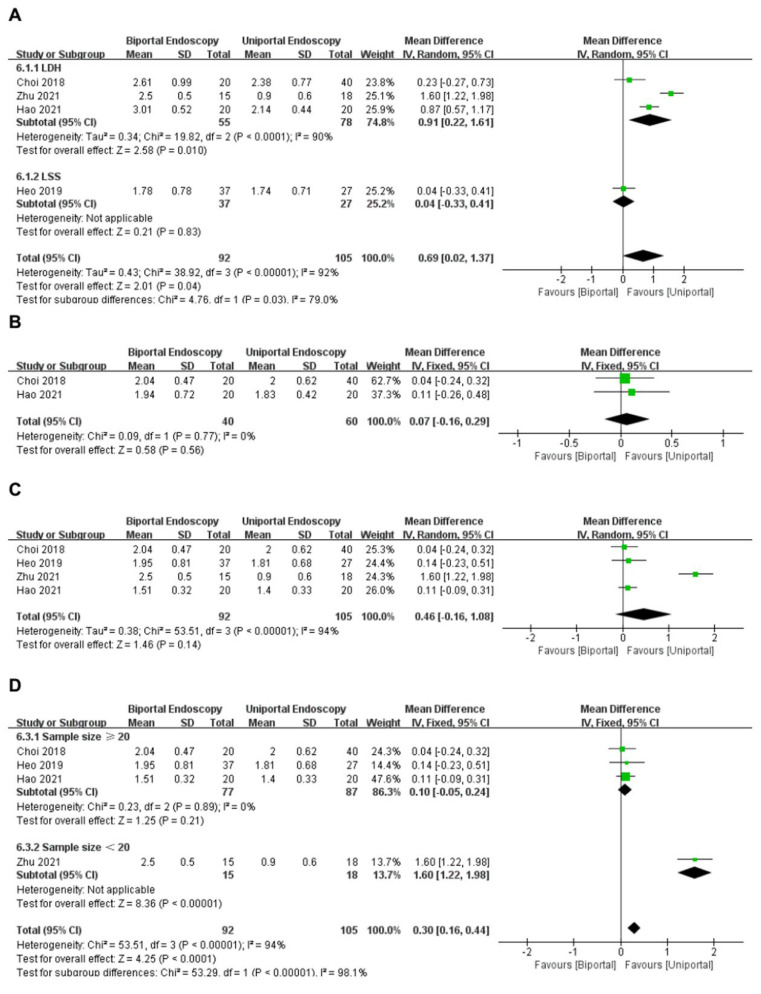
Forest plot of VAS back pain score (**A**) subgroup analysis within 3 days; (**B**) within 3 months; (**C**) last follow-up; (**D**) subgroup analysis at the last follow-up Choi et al., 2018 [15]; Heo et al., 2019 [16]; Zhu et al., 2021 [14]; Hao et al., 2021 [17].

**Figure 4 medicina-58-01523-f004:**
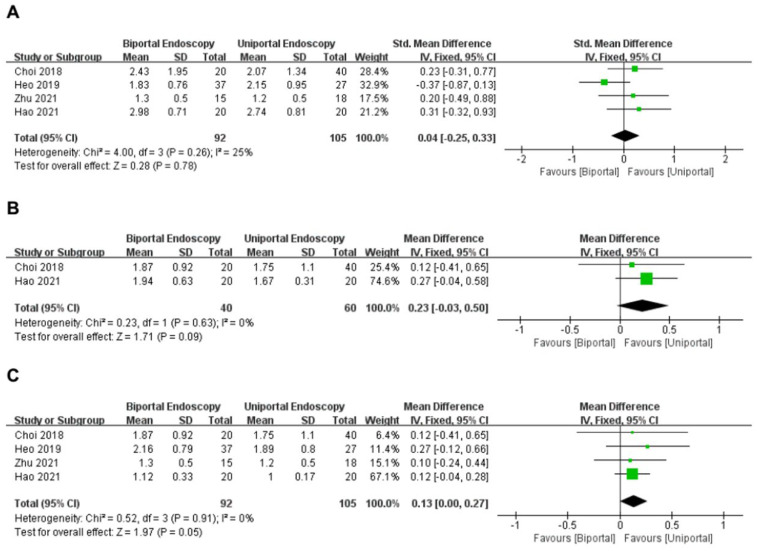
Forest plot of VAS leg pain score (**A**) within 3 days; (**B**) within 3 months; (**C**) last follow-up Choi et al., 2018 [15]; Heo et al., 2019 [16]; Zhu et al., 2021 [14]; Hao et al., 2021 [17].

**Figure 5 medicina-58-01523-f005:**
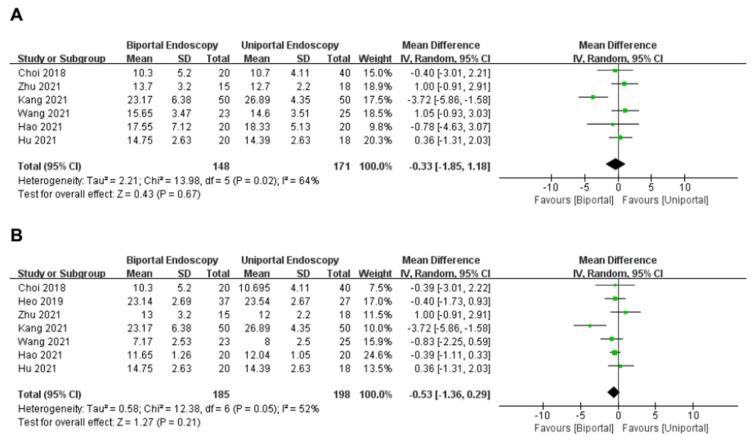
Forest plot of ODI (**A**) within 3 months; (**B**) last follow-up Choi et al., 2018 [15]; Heo et al., 2019 [16]; Zhu et al., 2021 [14]; Kang et al., 2021 [11]; Wang et al., 2021 [12]; Hao et al., 2021 [17]; Hu et al., 2021 [13].

**Figure 6 medicina-58-01523-f006:**
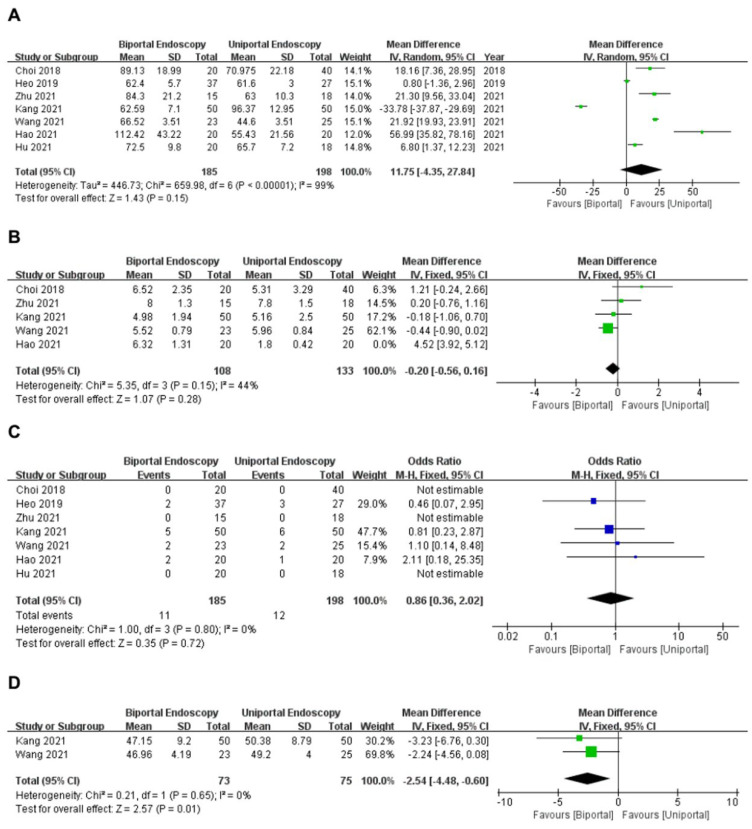
Forest plot (**A**) operation time; (**B**) length of hospital stay; (**C**) complication; (**D**) intraoperative estimated blood loss Choi et al., 2018 [15]; Heo et al., 2019 [16]; Zhu et al., 2021 [14]; Kang et al., 2021 [11]; Wang et al., 2021 [12]; Hao et al., 2021 [17]; Hu et al., 2021 [13].

**Figure 7 medicina-58-01523-f007:**
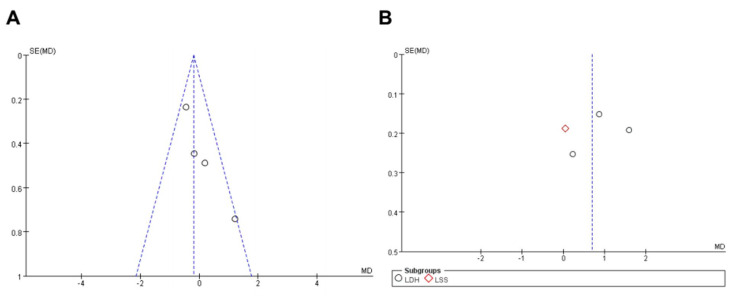
Funnel chart of publication bias (**A**) length of hospital stay; (**B**) VAS for back (within 3 days).

**Table 1 medicina-58-01523-t001:** Characteristics and Primary Outcomes of Included Studies.

Study (Year)	Country Study Type	Patient	Surgery Procedures	Sample Size	Age (Mean ± SD)	Gender (M/F)	Follow-Up (m)	Outcomes
Choi et al., 2018 [15]	Korea prospective	LDH	Biportal	20	47.43 ± 12.21	10/10	1	①③⑤⑥
			Uniportal	40	44.45 ± 7.87	20/20	1	
Heo et al., 2019 [16]	Korea Retrospective	LSS	Biportal	37	66.7 ± 9.4	15/22	12.5 ± 3.3	①④⑤⑥
			Uniportal	27	67.3 ± 9.9	11/16	12.5 ± 3.3	
Zhu et al., 2021 [14]	China Retrospective	LDH	Biportal	15	54 (Median)	7/8	6–18	①③⑤⑥
			Uniportal	18	56 (Median)	11/7	6–19	
Kang et al., 2021 [11]	China Retrospective	LSS	Biportal	50	64.97 ± 9.83	22/28	3	①②③④⑤⑥
			Uniportal	50	65.18 ± 11.12	26/24	3	
Wang et al., 2021 [12]	China Retrospective	LSS	Biportal	23	61.52 ± 4.09	13/10	9.26 ± 0.75	①②③④⑤⑥
			Uniportal	25	59.24 ± 4.11	13/12	8.96 ± 0.89	
Hao et al., 2021 [17]	China Retrospective	LDH	Biportal	20	58.2 ± 10.2	14/6	6 (at least)	①②③④⑤⑥
			Uniportal	20	59.3 ± 7.8	8/12	6 (at least)	
Hu et al., 2021 [13]	China RCT	LDH	Biportal	20	59.3	25	3	①⑤⑥
			Uniportal	18		13	3	

RCT, randomized controlled trial; LDH, lumbar disc herniation; LSS, lumbar spinal stenosis; ① Operation time (min); ② Intraoperative estimated blood loss (ml); ③ Length of hospital stay (days); ④ Complications; ⑤ ODI (%); ⑥ VAS score (back, leg pain).

**Table 2 medicina-58-01523-t002:** Quality Assessment of Included Studies Based on Newcastle–Ottawa Scale.

Study	Selection	Comparability	Outcome/Exposure	Quality Judgment
Choi et al., 2018 [15]	3	2	1	6
Heo et al., 2019 [16]	3	2	2	7
Zhu et al., 2021 [14]	3	2	2	7
Kang et al., 2021 [11]	3	2	1	6
Wang et al., 2021 [12]	3	2	3	8
Hao et al., 2021 [17]	4	2	2	8

The numbers (1–8) represent the number of NOS stars, with a maximum score of 9 stars.

**Table 3 medicina-58-01523-t003:** Complications of Included Studies.

Study (Year)	Surgery Procedures	Sample Size	No. of Complications								
			Dural Tear	Nerve Root Injury	Transient Weakness	Postop Hematoma	Postop Instability	Infection	Transient Paresthesia	Cerebrospinal Fluid Leakage/Headache	Total Complications
Choi et al., 2018 [15]	UBE	20	0	0	0	0	0	0	0	0	0
	PIED + PTED	40	0	0	0	0	0	0	0	0	0
Heo et al., 2019 [16]	UBE	37	1	0	0	1	0	0	0	0	2
	PIED	27	1	0	1	1	0	0	0	0	3
Zhu et al., 2021 [14]	UBE	15	0	0	0	0	0	0	0	0	0
	PIED	18	0	0	0	0	0	0	0	0	0
Kang et al., 2021 [11]	UBE	50	2	3	0	0	0	2	0	0	5
	PIED	50	3	2	0	0	0	4	0	0	6
Wang et al., 2021 [12]	UBE	23	1	0	0	0	0	0	1	0	2
	PIED	25	1	0	0	0	0	0	1	0	2
Hao et al., 2021 [17]	UBE	20	0	0	0	0	0	0	0	2	2
	PTED	20	0	0	0	0	0	0	1	0	1
Hu et al., 2021 [13]	UBE	20	0	0	0	0	0	0	0	0	0
	PIED	18	0	0	0	0	0	0	0	0	0

UBE, unilateral biportal endoscopic technique; PTED, percutaneous transforaminal endoscopic discectomy; PIED, percutaneous interlaminar endoscopic discectomy; +, and.

## Data Availability

The detailed data of this study are available from the corresponding.

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
