# Peer review of "Is Biportal Endoscopic Spine Surgery More Advantageous Than Uniportal for the Treatment of Lumbar Degenerative Disease? A Meta-Analysis"

_medicina, 2022, doi:10.3390/medicina58111523_

Round 1
Reviewer 1 Report
Dear Authors!
Thank you very much for sharing your thoughts about the uniportal and biportal endoscopic spine surgery of degenerative lumbar diseases.
However, the article presents several issues that should be addressed.
Why did you exclude patients with serious cardiovascular diseases, mental diseases etc.? How did you exclude them?
Why did you include only 7 studies, of those only 3 English and 4 Chinese?
Majority of Chinese studies have a sample size <25. How is it possible to perform strong statistics with such a small number? How does it affect your statistics as you had to exclude the study published by Hao et al due to the enormous heterogeneity?
What is new for the readers? What makes your article special? Every result was published before in literature.
What is the take home message of the article?
Author Response
Firstly, we would like to thank you for allowing me to revise and for the reviewers’ constructive comments concerning our article (Medicina-1878864). These comments are all valuable and helpful for improving our article. All the authors have seriously discussed all these comments. According to the reviewers’ comments, we have tried our best to modify our manuscript to meet the requirements of your journal. In this revised version, changes to our manuscript within the document were all highlighted by using yellow-coloured text. Point-by-point responses to the reviewers are listed below.
Reviewer 1
Q1. Why did you exclude patients with serious cardiovascular diseases, mental diseases etc.? How did you exclude them?
Answer: Thanks for your question. First of all, most of the original clinical studies excluded patients with serious cardiovascular and mental diseases. These patients may affect the accuracy of functional evaluation. Therefore, we screened the literature according to the exclusion criteria in the original study.
Q2. Why did you include only 7 studies, of those only 3 English and 4 Chinese?
Answer: Thanks for your question. Because biportal endoscopic spine surgery (BESS) is an emerging technique for spinal surgery, there are few studies comparing the clinical outcomes of uniportal and biportal full endoscopic spine surgery for the treatment of lumbar degenerative disease (LDD). Therefore, after our systematic search, only seven studies met our inclusion criteria.
Q3. Majority of Chinese studies have a sample size <25. How is it possible to perform strong statistics with such a small number? How does it affect your statistics as you had to exclude the study published by Hao et al due to the enormous heterogeneity?
Answer: Thanks for your question. At present, the BESS technique is still in its infancy of being performed, so the sample sizes of some studies are not large. This is also mentioned in the Discussion Section (Limitations and strengths). In addition, the subgroup analysis of the sample size (Line 198) was conducted to obtain relatively reliable conclusions.
Due to the large heterogeneity, we utilized sensitivity analysis to find the source of heterogeneity and eventually found that the study published by Hao et al had significant heterogeneity in multiple categories. Therefore, it is reasonable to exclude this study for more reliable conclusions.
Q4. What is new for the readers? What makes your article special? Every result was published before in literature.
Answer: Thanks for your question. Before our study, no meta-analysis provided evidence for comparing the two techniques to determine which is more clinically effective in the treatment of LDD. Therefore, we performed this meta-analysis to comprehensively evaluate the correlation between the two techniques and the prognosis of patients with LDD. The results will assist and guide clinical decision-making.
Q5. What is the take home message of the article?
Answer: Thanks for your question. According to our meta-analysis, patients who underwent the uniportal endoscopic procedure had more significant early postoperative back pain relief than those who underwent the biportal endoscopic procedure. Nevertheless, both surgical techniques are safe and effective.
Reviewer 2 Report
The Authors perfomed a very interesting work about the comparison between biportal and uniportal spinal endoscopy. The issue is of interest for the spinal surgeons. The methodology sounds credible. At conclusion, data support the assumption, that both surgical techniques are safe and effective. Moreover, patients who underwent the uniportal endoscopic procedure had more significant early postoperative back pain relief than those who underwent the biportal endoscopic procedure.
Author Response
Firstly, we would like to thank you for allowing me to revise and for the reviewers’ constructive comments concerning our article (Medicina-1878864). These comments are all valuable and helpful for improving our article. All the authors have seriously discussed all these comments. According to the reviewers’ comments, we have tried our best to modify our manuscript to meet the requirements of your journal. In this revised version, changes to our manuscript within the document were all highlighted by using yellow-coloured text. Point-by-point responses to the reviewers are listed below.
Reviewer 2
The Authors performed a very interesting work about the comparison between biportal and uniportal spinal endoscopy. The issue is of interest for the spinal surgeons. The methodology sounds credible. At conclusion, data support the assumption, that both surgical techniques are safe and effective. Moreover, patients who underwent the uniportal endoscopic procedure had more significant early postoperative back pain relief than those who underwent the biportal endoscopic procedure.
Answer: Thank you for your recognition of our research.
Reviewer 3 Report
The authors provide a well-structured meta-analysis on biportal endoscopic procedures for LDD compared to the uniportal technique. Apart from a careful review by a native English speaker I would suggest to implement selection criteria, in methods section, specifying what kind of LDH have been treated in the manuscripts selected for the meta-analysis and which kind of disc herniations have been eventually excluded. This is of crucial interest for the readers, helping in understanding for what kind of disc herniations both the endoscopic procedures may result advantageous. In this sense authors should refer to the LDH classification already reported in literature (Acta Neurochir (Wien). 2017 Jul;159(7):1273-1281. doi: 10.1007/s00701-017-3198-9. Epub 2017 May 22. PMID: 28534073) adding this data even in the related tables (table 1 and 3) and eventually discussing significant differences in this sense among the considered studies.
With the advocated integrations I would support the paper for publication.
Author Response
Firstly, we would like to thank you for allowing me to revise and for the reviewers’ constructive comments concerning our article (Medicina-1878864). These comments are all valuable and helpful for improving our article. All the authors have seriously discussed all these comments. According to the reviewers’ comments, we have tried our best to modify our manuscript to meet the requirements of your journal. In this revised version, changes to our manuscript within the document were all highlighted by using yellow-coloured text. Point-by-point responses to the reviewers are listed below.
Reviewer 3
The authors provide a well-structured meta-analysis on biportal endoscopic procedures for LDD compared to the uniportal technique. Apart from a careful review by a native English speaker I would suggest to implement selection criteria, in methods section, specifying what kind of LDH have been treated in the manuscripts selected for the meta-analysis and which kind of disc herniations have been eventually excluded. This is of crucial interest for the readers, helping in understanding for what kind of disc herniations both the endoscopic procedures may result advantageous. In this sense authors should refer to the LDH classification already reported in literature (Acta Neurochir (Wien). 2017 Jul;159(7):1273-1281. doi: 10.1007/s00701-017-3198-9. Epub 2017 May 22. PMID: 28534073) adding this data even in the related tables (table 1 and 3) and eventually discussing significant differences in this sense among the considered studies.
With the advocated integrations I would support the paper for publication.
Answer: Thank you for your kind comments. Strongly agree with your suggestion, but unfortunately, the type of LDH is not provided in the original publications.
Reviewer 4 Report
Article entitled „Does biportal endoscopic spine surgery more advantageous 2 than uniportal for the treatment of lumbar degenerative dis- 3 ease? A meta-analysis” is very interesting, easy to read with a clear message. My comments are as follows; 1. The degree of lumbar disc herniation (LDH) and lumbar spinal stenosis (LSS) in patients qualified for endoscopic treatment of the spine was not analyzed. Authors should refer to this in the discussion of whether the severity of the disease may influence the outcome of treatment 2. In figure 1, the sentence studies incuded in quantitative synthesis (7)” is repeated twice. 3. Table 1 - the contents are unevenly distributed and therefore difficult to read. The relatively young age of the analyzed patients, the average of 44 to 67 years, is noteworthy. It is not known what is the result of diseases at a more advanced age. Authors should refer to this issue in the discussion. 4. Why there are six works in table 2 and not the analyzed seven. 5. What do the numbers 1-8 in table 2 mean, maybe it is worth explaining it in the legend 6. In Figure 3, the legend - what is the difference between c- last follow up and d-subgroup analysis at the last follow up.Author Response
Firstly, we would like to thank you for allowing me to revise and for the reviewers’ constructive comments concerning our article (Medicina-1878864). These comments are all valuable and helpful for improving our article. All the authors have seriously discussed all these comments. According to the reviewers’ comments, we have tried our best to modify our manuscript to meet the requirements of your journal. In this revised version, changes to our manuscript within the document were all highlighted by using yellow-coloured text. Point-by-point responses to the reviewers are listed below.
Reviewer 4
Q1. The degree of lumbar disc herniation (LDH) and lumbar spinal stenosis (LSS) in patients qualified for endoscopic treatment of the spine was not analyzed. Authors should refer to this in the discussion of whether the severity of the disease may influence the outcome of treatment.
Answer: Thanks for your advice. We strongly agree with your suggestion, so this part has been mentioned in the Discussion section (yellow highlighted).
Q2. In figure 1, the sentence studies included in quantitative synthesis (7)” is repeated twice.
Answer: Thanks for your question. In figure 1, one is qualitative analysis, the other is quantitative analysis.
Q3. Table 1 - the contents are unevenly distributed and therefore difficult to read. The relatively young age of the analyzed patients, the average of 44 to 67 years, is noteworthy. It is not known what is the result of diseases at a more advanced age. Authors should refer to this issue in the discussion.
Answer: Thanks for your advice. Table 1 has been modified as requested (yellow highlighted). And the issue of age has been mentioned in the Limitations and strength section (yellow highlighted).
Q4. Why there are six works in table 2 and not the analyzed seven?
Answer: Thanks for your question. The quality of non-randomized articles (6 studies) was assessed using the Newcastle-Ottawa scale (NOS) (Table 2). And the quality of the randomized controlled study (1 study) was described in Figure 2. Therefore, altogether seven studies were included in this meta-analysis.
Q5. What do the numbers 1-8 in table 2 mean, maybe it is worth explaining it in the legend?
Answer: Thanks for your advice. The relevant content has been supplemented in the legend in Table 2 (yellow highlighted). As we all know, the Newcastle-Ottawa Scale (NOS) for assessing the quality of non-randomized studies in meta-analyses. And the NOS evaluates the quality of studies using the semi-quantitative principle of the star system, with a maximum score of 9 stars.
Q6. In Figure 3, the legend - what is the difference between c- last follow up and d-subgroup analysis at the last follow up.
Answer: Thanks for your question. First of all, the data in Figure 3C and Figure 3D are exactly the same. Secondly, the Figure 3D better illustrates the results of the subgroup analysis.
Round 2
Reviewer 1 Report
Dear Authors,
thank you for answering the comments. However, I still do not believe that the article is suitable for publication.